# GAZELLE: A MULTIMODAL LEARNING SYSTEM ROBUST TO MISSING MODALITIES

## ABSTRACT

Typical multimodal classification systems exhibit deteriorated performance if one or more modalities are missing at test time. In this work, we propose a robust multimodal classification system, namely `Gazelle`, which is less susceptible to missing modalities. It consists of a single-branch network sharing weights across multiple modalities to learn intermodal representations and introduces a novel training scheme featuring a modality switch mechanism over input embeddings extracted using modality-specific networks to maximise performance as well as robustness to missing modalities. Extensive experiments are performed on four challenging datasets including textual-visual (UPMC Food-101, Hateful Memes, Ferramenta) and audio-visual modalities (VoxCeleb1). `Gazelle` achieves superior performance when all modalities are present as as well as in the case of missing modalities compared to the existing state-of-the-art methods.

## 1 INTRODUCTION

Social media users often combine audio, video and text modalities to express their opinions (Moon et al., 2018). These modalities generally complement each other enriching the understanding of a particular task (Baltrušaitis et al., 2018). Different combinations of these modalities have been extensively studied to solve various tasks such as multimodal classification (Kiela et al., 2018; 2020), cross-modal retrieval (Wang et al., 2016), cross-modal verification (Nagrani et al., 2018b), multimodal named entity recognition (Moon et al., 2018), visual question answering (Anderson et al., 2018; Fukui et al., 2016), image captioning (Vinyals et al., 2015), semantic relatedness (Kiela & Bottou, 2014), and multimodal machine translation (Specia et al., 2016; Elliott et al., 2016). Multimodal modeling is challenging due to the difference in structure and representations of various modalities. The existing multimodal systems have commonly used neural network-based mappings to learn the joint representation of multiple modalities. For example, separate independent networks are leveraged to extract embeddings of each modality to learn joint representations in multi-branch networks (Wang et al., 2016; Faghri et al., 2018; Nagrani et al., 2018b;a; Saeed et al., 2022; Kim et al., 2018). Likewise, some recent multimodal systems have leveraged Transformers to learn joint representations using two-branch networks (Lu et al., 2019; Tan & Bansal, 2019; Kim et al., 2021). In these methods, the modular nature of the multi-branch networks is instrumental in developing various multimodal applications and have demonstrated remarkable performance (Arevalo et al., 2017; Gallo et al., 2017; Vielzeuf et al., 2018; Kiela et al., 2018; 2020; Kim et al., 2021). However, a limitation of these methods is that they require complete modalities, as in training data, to demonstrate good testing performance.

Multimodal data collected from the real-world are often imperfect due to missing modalities, resulting in a significantly deteriorated performance of the existing models (Ma et al., 2022; 2021; Lee et al., 2023; Wang et al., 2023). For example, as seen in Table 1, ViLT (Kim et al., 2021), a Transformer-based model, demonstrates a drop in performance of 28.3% when 30% of the text modality is present (i.e., 70% missing) at test time. Surprisingly, the performance is even lower than the ViLT trained and tested using individual image modality (unimodal) by a margin of 5.6%. This deteriorated performance renders multimodal classification training ineffective for real-world scenarios where missing modality may be encountered. The drop in performance may be attributed to the commonly used multi-branch design implementing attention layers for modality interaction. Such a design may learn weights in a way that the final performance is highly dependent on the correct combination of input modalities (Lu et al., 2019; Kim et al., 2018). One typical way of ad-

Table 1: Comparison of `Gazelle` with ViLT (Kim et al., 2021)* on UPMC Food-101 (Wang et al., 2015) dataset under different training and testing settings. Δ ↓ indicates performance deterioration due to missing modality at test time. *ViLT values are taken from Tab. 1 of (Ma et al., 2022). Best results in each setting are shown in bold.

| Dataset | Methods | Settings | Training | | Testing | | Accuracy | Δ ↓ |
|---|---|---|---|---|---|---|---|---|
| | | | Image | Text | Image | Text | | |
| UPMC Food-101 | ViLT | Complete Modalities | 100% | 100% | 100% | 100% | 91.9 | - |
| | | Missing Modality | 100% | 100% | 100% | 30% | 65.9 | 28.3% |
| | | Unimodal | 100% | 0% | 100% | 0% | 71.5 | - |
| | `Gazelle` | Complete Modalities | 100% | 100% | 100% | 100% | **94.6** | - |
| | | Missing Modality | 100% | 100% | 100% | 30% | **84.8** | **12.3%** |
| | | Unimodal | 100% | 0% | 100% | 0% | **81.7** | - |

dressing the missing modality problem is to use a generation process such as Generative Adversarial Networks (GANs) to complete the modalities (Zhao et al., 2021; Suo et al., 2019). However, such methods require extensive training of the generative models to construct the missing modalities. Several researchers utilize a dummy input to the model in case of a missing modality, which results in a deteriorated performance of the model (Ma et al., 2022).

In this work, we target this problem of robustness to the missing modalities by hypothesizing that learning a shared representation across different modalities enables a common continuous representation space (Firat et al., 2016; Dabre et al., 2020). Such an intermodal representation benefits in case of missing modality at test time. Motivated by this, we propose `Gazelle` that utilizes weight sharing across multiple modalities in a single-branch network to enable learning of inter-modal representations. `Gazelle` utilizes pre-trained embeddings of each modality and learns a joint representation using a novel modality switching mechanism to carry out the training. It outperforms state-of-the-art (SOTA) methods on several multimodal classification datasets as well as demonstrates superior robustness against missing modality at test time. For instance, as seen in Table 1, compared to the existing multimodal SOTA method, ViLT (Kim et al., 2021), resulting in a classification accuracy of 91.9% when both image and text modalities are completely available on the UPMC Food-101 (Wang et al., 2015) dataset. Under the same setting, our approach outperforms ViLT by achieving an accuracy of 94.6%. Additionally, when training and testing are carried out using only image modality, i.e., unimodal training, ViLT results in a classification score of 71.5% which is substantially lower than the 81.7% of our approach, highlighting the significance of `Gazelle` for unimodal as well as multimodal applications. In case of severely missing modality (i.e., only 30% of text modality available during testing), ViLT demonstrates an accuracy of 65.9% which is even lower than the 71.5% of its unimodal image-only training. In contrast, `Gazelle` demonstrates substantial robustness against missing modality by achieving 84.9% accuracy when only 30% of text modality is available, which is superior to the unimodal performance. Similar trends are observed across other multimodal classification datasets used to evaluate our approach. The key contributions of our work are as follows:

1. `Gazelle`: A multimodal classification system robust to missing modalities at test time.

2. A novel modality switching mechanism is proposed that enables weight sharing across multiple modalities in our single-branch network.

3. A wide range of experiments are performed on four datasets including image-text (UPMC Food-101 (Wang et al., 2015), Hateful Memes (Kiela et al., 2020) and Ferramenta (Gallo et al., 2017)) audio-visual (Voxceleb1 (Nagrani et al., 2017)) modalities. The proposed system exhibits SOTA performance when complete modalities are present. Similarly, in the case of missing modalities, our approach demonstrates superior robustness compared to existing SOTA methods.

## 2 RELATED WORK

Multiple modalities including text, image, video, and audio often contain complementary information about a common subject. The goal of multimodal learning is to leverage this complementary information across multiple modalities to improve the performance of various machine learning tasks such as classification, retrieval, or verification. Each multimodal task is different from the other, while the underlying objective remains the same: to learn joint representations across multiple

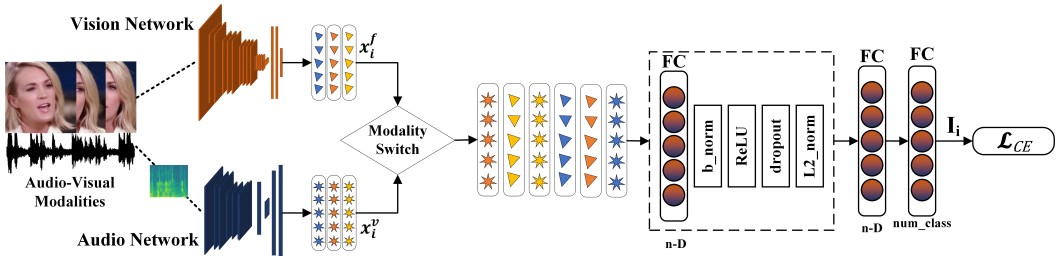

Figure 1: Architecture diagram of `Gazelle`. Modality-specific pre-trained networks (vision and audio networks in the given example) are used to extract embeddings which are passed through a modality switch and input to our single-branch network which learns modality independent representations to encode intermodal knowledge with weight sharing across multiple modalities.

modalities (Baltrušaitis et al., 2018). Existing multimodal methods employ multi-branch networks to learn joint representations by minimizing the distance between different modalities (Wang et al., 2016; Nagrani et al., 2018a; Saeed et al., 2022; Kim et al., 2018; Arevalo et al., 2017; Vielzeuf et al., 2018; Kiela et al., 2018; Muennighoff, 2020). Such methods using multi-branch networks have achieved remarkable performance (He & Peng, 2017; Wang et al., 2015; Gallo et al., 2020; Arevalo et al., 2017; Gallo et al., 2017; Nawaz et al., 2019; Vielzeuf et al., 2018; Kiela et al., 2018; Yang et al., 2019; Kiela et al., 2020). However, most multimodal systems suffer from performance deterioration if some modalities become absent at test time, a problem often referred to as handling missing modalities (Zhang et al., 2022a; Wang et al., 2022; Zhang et al., 2022b).

Considering the importance of multimodal systems, recent years have witnessed an increasing interest in handling the missing modality problem (Ma et al., 2021; 2022; Lee et al., 2023). Generally, existing multimodal methods that address this problem can be grouped into three categories. The first category is the input masking approach which randomly removes the inputs at training time to mimic missing modality information. For example, (Parthasarathy & Sundaram, 2020) introduced a strategy to randomly remove visual inputs during training to mimic missing modality scenarios for a multimodal emotion recognition task. The second category exploits the available modality to generate the missing one (Ma et al., 2021; Cai et al., 2018). For example, (Zhang et al., 2022b) generated the missing textual modality conditioned on the available visual modality. The third category learns a joint representation having related information from multiple modalities (Wang et al., 2020). For example, (Han et al., 2019) learned audio-visual joint representations to improve the performance of the unimodal emotion recognition task, however, it is not capable of exploiting complete modality information at the test time.

In contrast to the existing methods, we propose to learn intermodal representations with a single-branch network employing weight sharing across multiple modalities for training. Our proposed method not only demonstrates superior multimodal classification performance but also exhibits significant robustness towards missing modalities compared to the existing SOTA methods.

## 3 METHODOLOGY

In this section, we propose `Gazelle`, a multimodal classification system that is robust to missing modalities. It considers modality-specific embeddings extracted using pre-trained networks as input to a single-branch network. It employs a modality switch to jointly train the network with cross-entropy loss. `Gazelle` is built on the intuition that multiple embeddings extracted using modality-specific networks represent a similar concept but in a different representation space. The weight sharing using a single-branch network enables learning of intermodal representations of these concepts. The model then benefits from the intermodal representations when a modality is missing at inference time. Figure 1 presents our approach. In the following, we explain modality embedding extraction, modality switching, single-branch network, and loss formulation used to train our system.

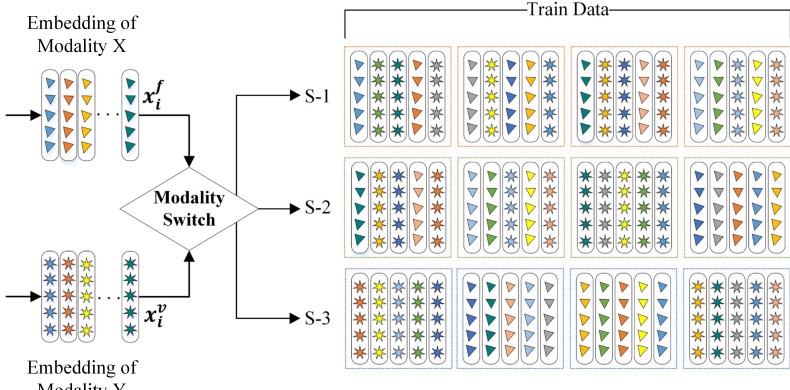

Figure 2: Our proposed modality switching strategies (S-1, S-2, S-3). For each of the three modality switching strategies, four batches are represented, each containing five embeddings for training single-branch network. S-1) all batches are multimodal in all epochs, S-2) half of the batches in each epoch are multimodal and the other half are unimodal, S-3) each batch has a different randomly selected modality.

## 3.1 PROBLEM FORMULATION

Given $\mathcal{D} = \{(x_i^f, x_i^v)\}_{i=1}^N$ is the training set where $N$ is the number of instances of pairs of modality $f$ and modality $v$ and $x_i^f$ and $x_i^v$ are individual modality embeddings of the $i^{th}$ instance, respectively. Moreover, each multimodal pair $(x_i^f, x_i^v)$ has a class label $y_i$. The embeddings are extracted using pre-trained modality-specific networks, such as Inception-ResNet-V1 (Szegedy et al., 2017), utterance level aggregator (Xie et al., 2019), and CLIP (Radford et al., 2021).

## 3.2 MODALITY SWITCHING

Typical multimodal learning systems take multiple modalities embeddings as input by using a multi-branch architecture (Nagrani et al., 2018a; Wang et al., 2016; Saeed et al., 2022). In contrast, Gazelle selects input embeddings in a sequential fashion. It is achieved by introducing a modality switching mechanism that determines the order in which embeddings are input to the single-branch network. This enables the network to map multiple modalities into a common but discriminative joint embedding space. Modality switching is critical to the training of our single-branch network. We propose and explore three modality switching mechanisms for training the network as shown in Figure 2:

S-1 Randomly selecting either of the available modalities resulting in a multimodal embedding stream at the output of the switch. In this strategy, all batches are multimodal while batch selection is also random.

S-2 In each epoch, 50% batches are multimodal as discussed in the first strategy, while the remaining 50% batches are unimodal. For each unimodal batch, either of the modality is randomly selected. During training, batch selection is random, resulting in a mixed stream of unimodal and multimodal batches.

S-3 In this strategy, all batches are unimodal. For each batch, either of the modality is randomly selected. During training, unimodal batches are then randomly selected, resulting in a multi-modal stream of unimodal batches.

We empirically evaluate the effectiveness of the three strategies and found S-1 to be the most effective. The results are presented in Section 5.2. Moreover, the experiments presented in the paper are based on S-1 strategy, unless stated otherwise.

## 3.3 NETWORK

As depicted in Figure 1, the network comprises of a single-branch of three blocks. The first block consists of a Fully Connected (FC) layer followed by Batch Normalization (BN), ReLU, and dropout

layers. The second block consists of an FC layer followed by $\ell_2$ normalization layers. The third block consists of an FC layer having the same size as the number of classes in a particular dataset followed by softmax. The weights of these FC layers are shared by different modality-specific embeddings which are input in a sequential fashion obtained from our modality switching mechanism. At test time, if complete modalities are present, late fusion is employed by taking the average of the logits obtained from the softmax layer over all modalities. In the case of only one modality, the fusion mechanism is not employed.

We employ cross entropy loss for training. Formally, we utilize a linear classifier with weights denoted as $\mathbf{W} = [\mathbf{w}_1, \mathbf{w}_2, ..., \mathbf{w}_C] \in \mathbb{R}^{d \times C}$ to compute the logits corresponding to $\mathbf{l}_i$ where $C$ is the number of classes and $d$ is the dimensionality of embeddings. The classification loss is then computed as:

$$\mathcal{L}_{CE} = -log \frac{exp(\mathbf{l}_i^T \mathbf{w}_{y_i})}{\sum_{j=1}^C exp(\mathbf{l}_i^T \mathbf{w}_j)} \tag{1}$$

## 4 EXPERIMENTS AND ANALYSIS

We evaluated `Gazelle` on the multimodal classification task using four datasets including textual-visual modalities based UPMC Food-101 (Wang et al., 2015), Hateful Memes (Kiela et al., 2021), Ferramenta (Gallo et al., 2017) and audio-visual modalities based Voxceleb1 (Nagrani et al., 2017). We conduct experiments using various settings including complete modalities and different levels of missing modalities. Moreover, an extensive ablation study is performed to evaluate different design choices of our approach. For a fair comparison, we adopt the same evaluation metrics used by the original authors of each dataset and the subsequent SOTA methods, i.e., classification accuracy and area under the receiver operating characteristic (AUROC).

### 4.1 DATASETS

Recently, Ma et al. (2022) introduces a comprehensive protocol to study missing modality problem on textual-visual data. To provide a comparison, we select the UPMC Food-101 dataset and the Hateful memes dataset from Ma et al. (2022). In addition, we select an audio-visual dataset (VoxCeleb1) to evaluate the generic applicability of `Gazelle` on other modalities. Finally, we select a widely popular and challenging multimodal dataset, Ferramenta, which is curated to resolve ambiguities among visual samples by using the textual modality.

**UPMC Food-**101. It is a classification dataset consisting of textual and visual modalities. The dataset was crawled from the web and each entry consists of an image and the HTML web page on which it was found. The dataset contains $90,704$ image-text pairs and 101 classes, and comes with a 75/25 train/test splits.
**Hateful Memes.** It is a multimodal dataset containing textual-visual pairs with binary labels and is developed with an aim to identify hate speech in memes. The dataset contains $10,000$ memes.
**Ferramenta.** It consists of $88,010$ textual-visual pairs belonging to 52 classes. The data is divided into $66,141$ instances for train and $2,186$ instances for test.
**Voxceleb**1. It is an audio-visual dataset of human speech videos extracted 'in the wild' from YouTube consisting of $1,251$ speakers. The data is divided into $145,265$ instances for train and $8,251$ instances for test.

### 4.2 IMPLEMENTATION DETAILS

**Network Settings.** `Gazelle` is trained using Adam optimizer with a learning rate of $0.01$ and dropout of $50\%$. The network has FC layers as: {input_dim, layer_dim, layer_dim, number of classes}, where the input_dim is 512 for audio-visual and 768 for textual-visual modalities. Moreover, layer_dim is 2048 for audio-visual and 768 for textual-visual modalities.

**Modality-Specific Embeddings.** We employ modality-specific networks to extract embeddings as explained in this section. Additional analysis by using other modality-specific extractors is also provided in Sec. 5.1 (Tab. 8). However, the following networks are our default experiment choices.

Table 2: Comparison of `Gazelle` with state-of-the-art multimodal methods on UPMC-Food-101. Best results are shown in bold; second best are underlined.

| Method | Accuracy |
|---|---|
| Wang et al. (2015) | 85.1 |
| Fused Representations (Nawaz et al., 2018) | 85.7 |
| MMBT (Kiela et al., 2019) | 92.1 |
| BERT+LSTM (Gallo et al., 2020) | 92.5 |
| Two-Branch (Saeed et al., 2022) | 94.2 |
| ViLT (Kim et al., 2021) | 91.9 |
| Ma et al. (2022) | 92.0 |
| Gazelle | **94.6** |

Table 3: Comparison of `Gazelle` with SOTA multimodal methods on Hateful Memes. *Results from Hateful Memes Challenge (Kiela et al., 2021). †Ensemble of 5 vision & language models. Best results are bold; second best are underlined.

| Method | AUROC |
|---|---|
| MMBT-Grid (Kiela et al., 2019)* | 67.3 |
| MMBT-Region (Kiela et al., 2019)* | 72.2 |
| ViLBERT (Lu et al., 2019)* | 73.4 |
| Visual BERT (Li et al., 2019)* | 73.2 |
| ViLT (Kim et al., 2021) | 70.2 |
| (Ma et al., 2022) | 71.8 |
| Vilio (Muennighoff, 2020)† | **82.5** |
| Gazelle | 72.5 |

Table 4: Comparison of `Gazelle` with state-of-the-art multimodal methods on Ferramenta dataset.

| Method | Accuracy |
|---|---|
| Ferramenta (Gallo et al., 2017) | 92.9 |
| Fused Representations (Nawaz et al., 2018) | 94.8 |
| IeTF (Gallo et al., 2018) | 95.2 |
| Two-Branch (Saeed et al., 2022) | 96.2 |
| MHFNet (Yue et al., 2023) | 96.5 |
| Gazelle | **96.5** |

Table 5: Comparison of `Gazelle` with state-of-the-art on VoxCeleb1 dataset.

| Method | AUROC |
|---|---|
| Two-branch (Saeed et al., 2022) | 97.7 |
| Gazelle | **98.0** |

**Image Embeddings** We extract image embeddings using Contrastive Language–Image Pre-training (CLIP) (Radford et al., 2021). The size of the output embeddings is 768, which matches with the corresponding text modality.

**Text Embeddings** We extract text embeddings from CLIP (Radford et al., 2021). The size of the output embedding is fixed to 768 to match the corresponding image modality.

**Face Embeddings** We extract face embeddings using Inception-ResNet-V1 (Szegedy et al., 2017) pre-trained with triplet loss (Schroff et al., 2015). The size of output embeddings is 512 which matches with the corresponding audio modality.

**Audio Embeddings** We extract audio embeddings using an utterance level aggregator (Xie et al., 2019) trained for a speaker recognition task with VoxCeleb1 (Nagrani et al., 2017) dataset. The size of output embeddings is kept 512 to match the corresponding face embeddings. The network is trained with a fixed-size spectrogram corresponding to a 2.5 second temporal segment, extracted randomly from each utterance (Xie et al., 2019).

### 4.3 EVALUATIONS UNDER COMPLETE MODALITIES SETTING

We first evaluate `Gazelle` when complete modalities are present during training and testing and compare the results with existing SOTA methods. Tables 2, 3, 4 and 5 present the results. `Gazelle` achieves SOTA performance on three out of four datasets. More specifically, on the UPMC-Food-101 dataset (Table 2), our model achieved a classification performance of 94.6%, outperforming all existing methods. On Ferramenta and VoxCeleb1 (Table 4 and 5) datasets, our model achieved 96.5% and 98.0% accuracy, respectively, outperforming all SOTA methods. Only on the Hateful Memes dataset (Table 3), `Gazelle` did not achieve SOTA results but demonstrated a comparable performance with several methods except Vilio (Muennighoff, 2020). It may be noted that Vilio is an ensemble method employing five vision and language models to achieve the reported AUROC, thus not directly comparable to any of the methods that are based on using a single system for inference.

Table 6: Evaluation of `Gazelle` with different levels of available modality in test set using UPMC-Food-101 and Hateful memes datasets. AUROC and accuracy are reported for Hateful Memes and UPMC-Food-101 respectively. Comparison is provided with Two-branch (Saeed et al., 2022), ViLT (Kim et al., 2021)*, and (Ma et al., 2022). *ViLT values are taken from (Ma et al., 2022). Boldface and underline denote, respectively, the best and second best results.

| Dataset | Training | | Testing | | `Gazelle` | Two-branch | ViLT | (Ma et al., 2022) |
|---|---|---|---|---|---|---|---|---|
| | Image | Text | Image | Text | | | | |
| UPMC Food-101 | 100% | 100% | 100% | 100% | **94.6** | 94.2 | 91.9 | 92.0 |
| | 100% | 100% | 100% | 90% | **93.2** | 93.1 | 88.2 | 90.5 |
| | 100% | 100% | 100% | 70% | **90.9** | 90.9 | 80.7 | 87.1 |
| | 100% | 100% | 100% | 50% | **88.2** | 87.8 | 73.3 | 82.6 |
| | 100% | 100% | 100% | 30% | **84.8** | 84.5 | 65.9 | 77.5 |
| | 100% | 100% | 100% | 10% | **83.3** | 81.6 | 58.4 | 73.3 |
| | 100% | 100% | 100% | 0% | **82.0** | 80.0 | - | - |
| | 100% | 0% | 100% | 0% | **81.7** | 81.7 | 71.5 | 71.5 |
| Hateful Memes | 100% | 100% | 100% | 100% | **72.5** | 61.1 | 70.2 | 71.8 |
| | 100% | 100% | 100% | 90% | **72.2** | 60.9 | 68.8 | 69.7 |
| | 100% | 100% | 100% | 70% | **72.0** | 60.2 | 65.9 | 66.6 |
| | 100% | 100% | 100% | 50% | **72.1** | 60.0 | 63.6 | 63.9 |
| | 100% | 100% | 100% | 30% | **71.3** | 59.4 | 60.2 | 61.2 |
| | 100% | 100% | 100% | 10% | **71.2** | 59.7 | 58.0 | 59.6 |
| | 100% | 100% | 100% | 0% | **71.2** | 59.5 | 54.9 | - |
| | 100% | 0% | 100% | 0% | **68.2** | **68.2** | 56.3 | 56.3 |

Table 7: Classification accuracy of `Gazelle` on the configuration of 100% missing modality in test set using Ferramenta and VoxCeleb1 datasets. Comparison of our approach is provided with the Two-branch Network (Saeed et al., 2022) to understand the importance of our single branch design. Best results are printed in boldface.

| Dataset | Training | | Testing | | `Gazelle` | Two-branch |
|---|---|---|---|---|---|---|
| | Image | Text | Image | Text | | |
| Ferramenta | 100% | 100% | 100% | 100% | **96.5** | 96.2 |
| | 100% | 100% | 100% | 0% | **92.3** | 71.0 |
| | 100% | 100% | 0% | 100% | **93.4** | 61.6 |
| | 100% | 0% | 100% | 0% | **92.5** | 92.5 |
| | Image | Audio | Image | Audio | | |
| VoxCeleb1 | 100% | 100% | 100% | 100% | **98.0** | 97.7 |
| | 100% | 100% | 100% | 0% | **84.7** | 38.9 |
| | 100% | 100% | 0% | 100% | **82.4** | 31.5 |
| | 100% | 0% | 100% | 0% | **84.2** | 84.2 |

## 4.4 EVALUATIONS UNDER MISSING MODALITIES SETTING

Ma et al. (2022) have shown that multimodal methods are brittle to missing modalities at test time. `Gazelle` aims to show better robustness towards missing modalities by learning intermodal representations. Table 6 compares our approach with existing SOTA methods; ViLT (Kim et al., 2021), Ma et al. (2022), and Two-Branch Network (Saeed et al., 2022) for varying amounts of missing modality on UPMC Food-101 and Hateful Memes datasets. As seen, our approach outperformed all existing SOTA methods with considerable margins. In the case of severely missing text modality (when only 10% is available), on the UPMC Food-101 dataset, our approach demonstrates an accuracy of 83.3% compared to 94.6% with 100% availability of all modalities. Compared to this, Two-branch Network, ViLT and Ma et al. (2022) demonstrate performances of 81.6, 58.4, and 73.3, respectively. Similarly, on the Hateful Memes dataset, `Gazelle` demonstrates an AUROC of 71.2% when only 10% of text modality is available at test time. In comparison, Two-branch Network, ViLT, and Ma et al. (2022) demonstrate performances of 59.7, 58.0, and 59.6, respectively. Similar trends are observed on Ferramenta and VoxCeleb1 datasets, as seen in Table 7, where comparisons are provided with the Two-branch Network (Saeed et al., 2022). This demonstrates the significance of our proposed single-branch network for multimodal training robust to missing modalities. Our work serves as a proof of concept that encourages researchers to consider weight sharing in building robust state-of-the-art multimodal networks.

Table 8: Performance comparison of `Gazelle` with the extracted embeddings using various pre-trained models. Best results are obtained when using CLIP as an image and text feature extractor.

| Dataset | Image Emb. | Text Emb. | Emb Size | Training | | Testing | | Accuracy |
|---|---|---|---|---|---|---|---|---|
| | | | | Image | Text | Image | Text | |
| UPMC Food-101 | ResNet-101 | Doc2Vec | 2048 | 100% | 100% | 100% | 100% | 87.5 |
| | ViT | Doc2Vec | 768 | 100% | 100% | 100% | 100% | 88.6 |
| | CLIP | Doc2Vec | 768 | 100% | 100% | 100% | 100% | 92.2 |
| | ViT | CLIP | 768 | 100% | 100% | 100% | 100% | 92.8 |
| | CLIP | CLIP | 768 | 100% | 100% | 100% | 100% | **94.6** |

Table 9: Performance analysis of the modality switching strategies (Figure 2). Results are reported using 100% modalities in the training set. $\Delta \downarrow$ indicates performance deterioration due to missing modalities at test time. Best results are shown in boldface.

| Dataset | Strategy | Testing | | Accuracy | $\Delta \downarrow$ |
|---|---|---|---|---|---|
| | | Image | Text | | |
| UPMC Food-101 | S-1 | 100% | 100% | **94.6** | - |
| | | 100% | 0% | **82.0** | **13.3%** |
| | S-2 | 100% | 100% | 94.5 | - |
| | | 100% | 0% | 81.1 | 14.3% |
| | S-3 | 100% | 100% | 93.2 | - |
| | | 100% | 0% | 81.7 | 13.6% |

## 5 ANALYSIS AND DISCUSSION

In this section, we provide further analysis on the impact of various embedding extractors and different modality switching strategies on the training and robustness of `Gazelle`.

### 5.1 EMBEDDING EXTRACTORS

In order to explore the optimal embedding extractor that enables the learning of common semantics, we carry out experiments using different pre-trained feature extractors including ResNet-101 (He et al., 2016), ViT (Dosovitskiy et al., 2020), and CLIP (Radford et al., 2021) for image embeddings and Doc2Vec (Le & Mikolov, 2014) and CLIP (Radford et al., 2021) for text embeddings. For a fair comparison, the experiments were conducted with complete modalities available during training and testing. As seen in Table 8, on the UPMC Food-101 dataset, the best performance is achieved when CLIP features are used for both image and text modalities. Therefore, unless stated otherwise, all of our experiments on image and text modalities are conducted using features extracted through CLIP.

### 5.2 MODALITY SWITCHING STRATEGIES

We evaluate the impact of various switching strategies on the proposed multimodal approach. Table 9 compares the results on the UPMC Food-101 dataset. `S-1`, where all batches in an epoch are multimodal, resulted in the best performance over the three strategies studied in this section by demonstrating 94.6% accuracy and a drop of 13.3% when the text modality is 100% missing during testing. On the other hand, `S-2`, where half of the batches in an epoch are multimodal and the other half are unimodal, resulted in a slightly lower accuracy of 94.5% and a drop of 14.3% when the text modality is completely missing. `S-3`, where all batches in an epoch are unimodal, demonstrates the lowest performance with only 93.2% accuracy on complete modalities and drops by 13.6% when the text modality is entirely missing. Therefore, the results reported in our manuscript use the `S-1` training strategy unless otherwise mentioned.

### 5.3 QUALITATIVE RESULTS

In addition to the empirical results and analysis, we use t-SNE to visualize the embedding space of `Gazelle` with complete modalities as well as missing textual modality on UPMC Food-101. The visualizations are helpful in observing the overall effect of our proposed training. Figure 3a shows t-SNE visualization of the embedding space extracted from modality-specific network (CLIP). Al-

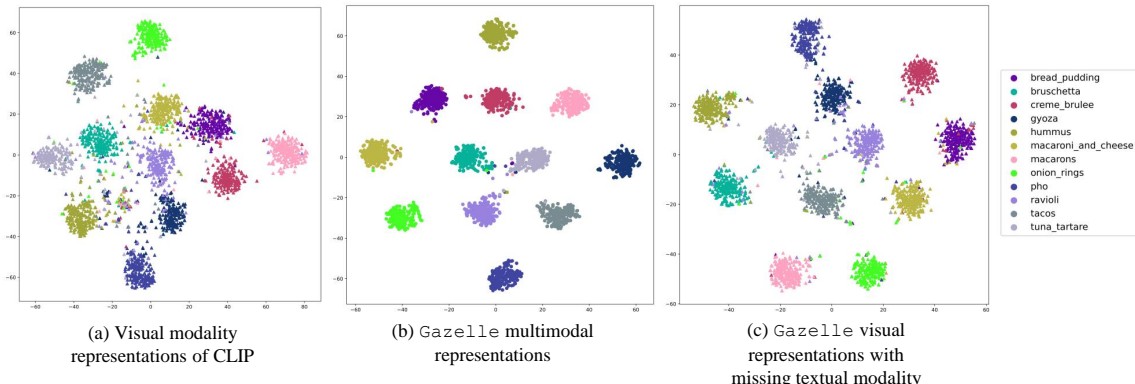

(a) Visual modality
representations of CLIP

(b) `Gazelle` multimodal
representations

(c) `Gazelle` visual
representations with
missing textual modality

Figure 3: t-SNE visualizations of (a) CLIP visual modality representations and (b,c) the embedding space of `Gazelle` (embeddings from the second block) on test set of UPMC Food-101. It can be seen that `Gazelle` not only enhances the classification boundaries when complete modalities are available at test time but also retains these boundaries when the textual modality is completely missing during test time. More on this and t-SNE visualization comparisons with the existing SOTA methods are provided in the supplementary.

though several classes are separable (highlighting the reasonable quality of the extracted embeddings), some overlap among the classes is observable. Figure 3b shows the multimodal embeddings extracted from the second block of trained `Gazelle`. It can be seen that `Gazelle` improves the overall classification boundaries highlighting the success of the proposed multimodal training. Finally, Figure 3c shows the embeddings extracted from `Gazelle` when textual modality is completely missing at test time. Although some distortions are noticeable, the overall separability of the classes is retained. This demonstrates the robustness of `Gazelle`, our proposed multimodal learning approach, towards missing modalities. An extensive version of Figure 3 is provided in the supplementary where we also compare additional SOTA method visualizations for comparison.

# 6 ADVANTAGES AND LIMITATIONS

**Advantages:** Notable advantages of our approach are high performance and robustness to missing modality. Moreover, `Gazelle` includes a significantly smaller network that results in fewer parameters for training. For example, compared to 2.44 million parameters of Two-branch Network (Saeed et al., 2022), our architecture requires merely 1.26 million parameters when both are trained on image and text modalities. This makes `Gazelle` easily trainable compared to Transformers or other complex attention-based mechanisms (Ma et al., 2022; Kim et al., 2021).

**Limitations:** `Gazelle` utilizes a modality switching mechanism leveraging weight sharing across multiple modalities with a single-branch network. Such a design requires extracted representations from modality-specific networks to have the same embedding size. We can perform a transformation of the embedding to the required size. However, this requires further experimentation and is out of the scope of our work.

# 7 CONCLUSION

We proposed a multimodal learning system that is substantially more robust against missing modalities compared to existing methods. To ensure robustness, a modality switching mechanism is proposed to serialize the embedding streams of single-branch networks. It facilitates complete weight sharing of the network across multiple modalities and encodes the shared semantics across modalities. Extensive experiments are performed on four datasets including audio-visual (VoxCeleb1) and textual-visual modalities (UPMC Food-101, Hateful Memes, and Ferramenta). The proposed system is thoroughly evaluated for missing modalities. The performance drop of the proposed system is noticeably smaller than the existing methods showing significant robustness. Excellent results demonstrate the proposed system as a new paradigm for multimodal learning systems robust to the loss of modalities.

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
