# GAZELLE: A MULTIMODAL LEARNING SYSTEM ROBUST TO MISSING MODALITIES

## A APPENDIX

### A.1 t-SNE VISUALIZATIONS

In addition to the empirical results and analysis provided in the main manuscript, we use t-SNE to visualize and compare the embedding space of `Gazelle` and Two-branch Network with complete modalities as well as missing modalities. The visualizations are helpful in observing the overall effect of our proposed training. Figures 4a & 4b show t-SNE visualizations on UPMC Food-101 dataset where embeddings of each modality are plotted directly before passing them into `Gazelle` and Two-branch Network. Although several classes in each modality are separable (highlighting the reasonable quality of the extracted modality-specific embeddings), some overlap among the classes is observable. Figure 4c & 4d show the t-SNE visualization of the features extracted from the second block of `Gazelle` and Two-branch Network. It can be observed that classes are more separable in the embedding space extracted from `Gazelle` (Figure 4c) compared to the Two-branch Network (Figure 4d) as well as modality-specific network (Figures 4a & 4b), demonstrating that our single-branch network successfully learns joint representations. Furthermore, the t-SNE visualizations in Figure 4e, 4f, 4g & 4h show that when one of the two modalities are missing during testing the two-branch network shows high distortions. Especially, when the visual modality is missing, the Two-branch Networks fails to separate most of the classes. In the case of `Gazelle`, although some distortion is noticeable when either of the modalities is missing, the overall separability of most of the classes is retained. This demonstrates the robustness of `Gazelle`, our proposed multimodal learning approach towards missing modalities.

### A.2 ROBUSTNESS TO CORRUPT MODALITIES

In real-world scenarios, it is often possible that a modality is not missing but corrupt due to several reasons including faulty equipment, low bandwidth, etc. In order to evaluate the robustness of `Gazelle` on corrupt modality, we performed a series of experiments by adding different level of noise to the test features of both modalities and report the results in Table 10. As seen, `Gazelle` shows reasonable tolerance to 100% corrupt modalities with Gaussian noise ($\sigma = 0.1, 0.5$) at test time. Under extreme case ($\sigma = 1.0$), the performance however deteriorates noticeably.

### A.3 MULTIMODAL TRAINING VS. UNIMODAL TRAINING

Generally, multimodal systems are popular because of their performance improvements over the models trained on individual modalities. However, when a multimodal system is exposed to missing modalities at test time, the performance often drops lower than that of the models trained on individual modalities, voiding the benefit of multimodal training (**?**). Therefore, we performed series of experiments to observe whether our approach demonstrates better performance in case of missing modality compared with the individual modality training. To carry out these experiments, instances from the test data are randomly eliminated to evaluate the robustness of the proposed method on missing modalities. Figure 5 shows that our approach demonstrates less susceptibility as the percentage of missing data increases for either of the modalities. Notably, compared to the models trained and tested on individual modalities, our approach retains better performance even when one of the modalities is missing over 90%. For example, our method produces 96.5% classification accuracy when both modalities are available on Ferramenta dataset. When visual modality is completely missing, the accuracy performance is 93.4% which is higher than the model trained on the individ-

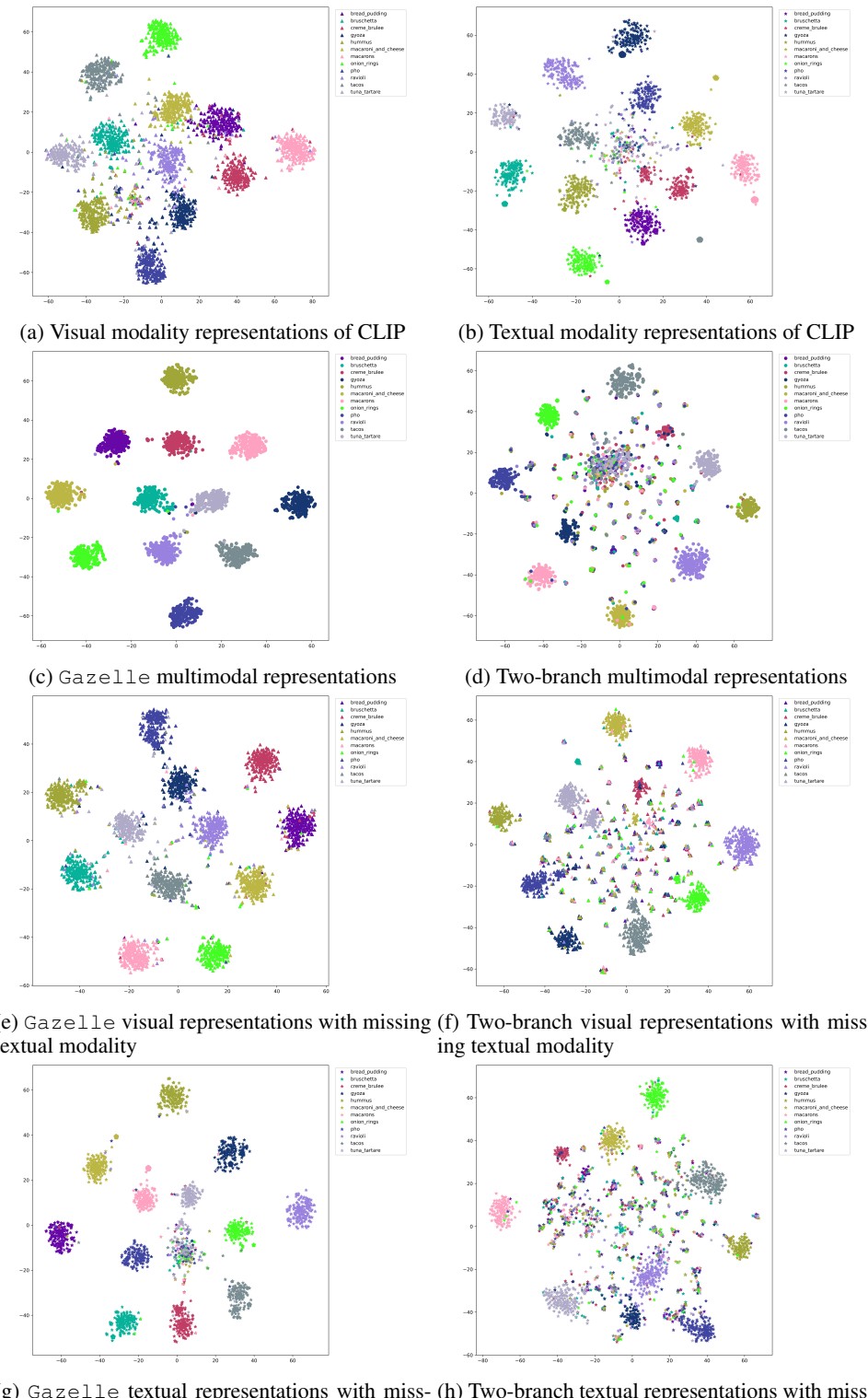

(a) Visual modality representations of CLIP

(b) Textual modality representations of CLIP

(c) Gazelle multimodal representations

(d) Two-branch multimodal representations

(e) Gazelle visual representations with missing textual modality

(f) Two-branch visual representations with missing textual modality

(g) Gazelle textual representations with missing visual modality

(h) Two-branch textual representations with missing visual modality

Figure 4: t-SNE visualizations of Gazelle and Two-branch Network (?) on UPMC Food-101 dataset. Note that the visualizations in 4c 4d ,4e,4f, 4g, & 4h are plotted with features from the second block of Gazelle and Two-branch Network.

Table 10: The performance of our approach with $100\%$ corrupted modalities by adding Gaussian noise. The standard deviation ($\sigma$) is varied whereas mean ($\mu$) is set to 0. $\Delta \downarrow$ indicates percentage of performance deterioration due to noise in modalities. Results are reported on accuracy with best results shown in boldface.

| Dataset | $\sigma$ | Training | | Testing | | Results | $\Delta \downarrow$ |
|---|---|---|---|---|---|---|---|
| | | Image | Text | Image | Text | | |
| UPMC Food-101 | 0.0 | 100% | 100% | 100% | 100% | **94.6** | - |
| | 0.1 | 100% | 100% | 100% | 100% | 94.4 | 0.2% |
| | 0.5 | 100% | 100% | 100% | 100% | 86.8 | 8.1% |
| | 1.0 | 100% | 100% | 100% | 100% | 49.8 | 47.3% |

ual modality ($92.5\%$). This demonstrates the significance of our proposed single-branch approach in multimodal representation learning.

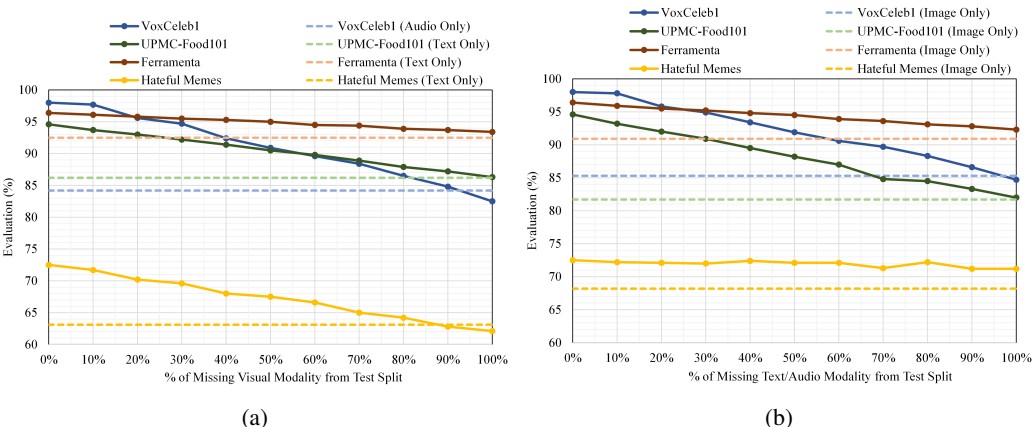

(a)                                                    (b)

Figure 5: Performance evaluation of `Gazelle` on missing modalities over four datasets including textual-visual (UPMC Food-101, Hateful Memes, Ferramenta) and audio-visual modalities (VoxCeleb1). Dotted lines represent unimodal results. (a) Visual modality is gradually dropped from 0% to 100% by randomly removing samples from the test data. (b) Audio modality (in case of VoxCeleb1) and text modality (in case of other three datasets) is gradually dropped from 0% to 100% by randomly eliminating samples from the test data.