# OpenReview forum: "Gazelle: A Multimodal Learning System Robust to Missing Modalities"
_ICLR.cc/2024/Conference — ICLR 2024 Conference Withdrawn Submission_

### Official Review · Reviewer_1WPw · 2023-10-30

**Soundness:** 1 poor
**Presentation:** 2 fair
**Contribution:** 2 fair
**Rating:** 3
**Confidence:** 4

**Summary:**

This paper proposes a multi-modal training method to enhance the robustness of multi-modal models to modality missing. Furthermore, since this paper utilizes CLIP embeddings, its absolute performance is significantly superior to previous methods in some datasets.

**Strengths:**

1, Using CLIP embeddings has led to a significant improvement in the overall performance of the multi-modal model.

2, The proposed training method indeed makes the multi-modal model more robust in cases of missing modalities compared to conventional models.

However, I believe that many of the experiments in this paper are, in fact, unfair in their comparisons. I have provided a detailed explanation of this in the "Weaknesses" section.

**Weaknesses:**

1, The reason this multi-modal model can achieve SOTA results on several datasets is fundamentally due to the use of embeddings from pre-trained models (such as CLIP embeddings), rather than the inherent superiority of the proposed training method itself. If you want to demonstrate how good your proposed training method is, different training methods should be applied with the same backbones. For the reasons mentioned above, I find the significance of Tables 2, 3, 4, and 5 to be quite limited because the performance improvement is not a result of your paper's new method but rather the utilization of pre-trained models from previous works.

2, In Table 6, when comparing the proposed method with Ma et al., I believe there is a significant misconception here. You used the CLIP model pre-trained on a large-scale text-image dataset by OpenAI, while Ma et al. used the ViLT backbone. The absolute performance of the model in this paper is better than Ma et al., which may be due to the superiority of CLIP over ViLT, rather than the training method proposed in this paper is better than Ma et al.'s method. **A more accurate comparison should be based on the proportion of performance degradation.**   Specifically, when 10% of the text is missing, Gazelle shows a decrease of (94.6-93.2)/(94.6-81.7)=10.85%, while Ma et al. exhibits a decrease of (92.0-90.5)/(92.0-71.5)=7.32%. From this perspective, when 10% of the text is absent, Ma et al. experience a relatively smaller proportion of decrease. Your higher absolute performance is simply due to the use of stronger pre-trained model embeddings, not because your proposed method is superior.

3, The results in Table 6 for Hateful meme, where having 50% text performs better than having 70% text, and where 0% text and 10% text yield the same performance, are indeed puzzling. This could suggest that the method proposed in this paper may not make optimal use of the available text data.

4, The method proposed in this paper requires that the sizes of features from different modalities remain consistent, which actually limits the flexibility of the entire model. For example, it may prevent the combination of BERT-Large and ViT-B.

**Questions:**

See weaknesses.

---

> ### Author Response · Authors · 2023-11-17
> **Rebuttal - 1WPw**
>
> **W1 (SOTA performances)**.  It is partly correct that the strength of the embedding used will have a direct impact on the performance of our network. However, the robustness to missing modality is due to the single branch network with weight sharing which is the main contribution of our work (Tables 6 and 7).
>
> In addition, Unimodal image only performance with CLIP on UPMC Food-101 is 81.7. Whereas, multimodal performance with Gazelle is 94.6 which is 12.9% higher than CLIP unimodal image only. Similarly,  unimodal image-only performance with CLIP on Hateful Memes is 68.2. Whereas, multimodal performance with Gazelle is 72.5 which is 4.3% higher than CLIP unimodal image only. This clearly establishes the performance improvements our approach yields on top of a pre-trained CLIP embeddings.
>
> Furthermore, the Two-branch model in Tables 6 and 7 is by design a baseline two-branch model that is trained on the same CLIP features without our proposed weight sharing and training. As seen, our approach consistently outperforms the two-branch model in all settings/scenarios. While the selection of features does play a part (we analyzed it in Table 8), the comparison with the Two-branch network clearly highlights that CLIP features are not the sole reason for performance improvement in our proposed approach.
>
> **W2 (Percentage deterioration - Evaluation metric)**.  Can the reviewer provide any reference for the evaluation metric they are suggesting to evaluate the missing modality scenarios?
>
> As for us, we follow the exact same \% drop evaluation metric as used by Ma et al. to measure performance deterioration in case of missing modality. As seen in Ma et al., $\%$ drop in performance can be measured as: $\Delta \downarrow$  $=(P_{complete}-P_{missing})/P_{complete}$, where $P_{complete}$ and $P_{missing}$ are the performances over complete and missing modalities.
>
> In addition, let's suppose we have two deployed multimodal classification networks. Classification performances of network **a** and **b**  are 90.0 and 80.0 respectively when both modalities are present. In case of a missing modality, the classification performances of network **a**  and **b**  are 83.0 and 78.0 respectively. Though network **b** is more robust, its performance is still lower than network **a** and wouldn't be preferred for deployment.
>
> ***W3 (Variation in Hateful Memes results)**. We suggest this is due to the nature of the Hateful Memes dataset. It is a binary multimodal classification dataset with imbalance classes which equates to small variation in the results when modality is missed randomly between two experiments. Similar observations are common in missing modality literature, for example, Figure 3 (in the middle summarizing Hateful Meme dataset results) in Supplementary material of Lee et al. 2023.
>
> ***W4 (Limitations)**.  Gazelle utilizes a modality switching mechanism leveraging weight sharing across multiple modalities with a single-branch network. Such a design requires extracted representations from modality-specific networks to have the same embedding size. We can perform a transformation of the embedding to the required size. However, this requires further experimentation and is out of the scope of our work.

---

### Official Review · Reviewer_q4t6 · 2023-10-31

**Soundness:** 2 fair
**Presentation:** 2 fair
**Contribution:** 1 poor
**Rating:** 3
**Confidence:** 5

**Summary:**

The paper proposes a new method for multimodal learning while dealing with missing modalities. The proposed method uses a single-branch network and a modality switching mechanism that shares weights for multiple modalities.

**Strengths:**

The paper tackles the interesting and important problem of learning multimodal data while being able to deal with missing modalities.

**Weaknesses:**

There are a number of shortcomings in the paper:

- The writing is generally ok, but a bit concise imo. Starting off the introduction with "social media users" is a bit strange, given that multimodal data have far wider uses other than social media.

- The method section is unclear and not well-written. First, it states "...sequential fashion. It is achieved by introducing a modality switching mechanism that determines the order in which embeddings are input to the single-branch network." What are the theoretical foundations for this? why is this used? what is the motivation and intuition behind it? Next, the paper states that they have three possible strategies: 1- randomly switching, 2- swishing between multimodal and unimodal 50-50, 3- going only with unimodal. Yet, no details are provided. Which of these are the proposed method? Is the paper simply exploring three options? Are there no other options? why not set the ratio as a hyperparameter and optimize it?

- The entire method is basically explained in a single paragraph, making it almost impossible to understand the details, fundamental theories and motivations behind things, etc.

- The methods used for comparison in Tables 2 through 5 have many important papers missing.

- Especially for the missing modality experiments, only 1 comparison is done (against Ma et al., 2022). Unfortunately, this is not enough, even if the method was sound and explained properly. Further experiments are required to validate the method.

**Questions:**

Please see my comments under weaknesses.

---

> ### Author Response · Authors · 2023-11-17
> **Rebuttal - q4t6**
>
> **W1 (Starting off the introduction with )**. We can rephrase the first sentence in the introduction with "Recent years have seen a surge in the usage of multimodal data in various applications. For instance, users combine text, image, audio or video modalities to sell a product over an e-commerce platform or express views on social media platforms.""
>
>  **W2 (Important references missing)**. Both suggested references are cited in our submission. For the record, the work ''Are multimodal transformers robust to missing modality?'' is referenced at least  **17** times in our manuscript. Therefore, we humbly suggest the reviewer to read our submission again.
>
>  **W3 (Method section)**.  **Intuition** In existing multimodal methods, the performance deterioration in the case of missing modalities is attributed to the commonly used multi-branch design.  Such a design may learn weights in a way that the final performance is highly dependent on the correct combination of input modalities.
> Thus, we propose to input one modality to the network at a time making the network independent of the complete modality combination. To achieve this goal, we design a modality switching mechanism that selects input sequence to the network.
>
> We explore three possible modality switching strategies to study their effect on complete as well as missing modalities. The results reported in our manuscript use the S-1 training strategy unless otherwise mentioned. We have provided details of modality switching in Section 3.2 along with a Figure.
>
> **W3 (Entire method is basically explained in a single paragraph)**.   There are three Subsections of our proposed approach. In Subsection 3.1, we formally provide details on the input data, Subsection 3.2 provides details on modality switch. Finally, Subsection 3.3 provide details on the single-branch network. We highly appreciate it if the reviewer could pinpoint the ambiguities related to the details of our proposed method.
>
> **W4 (Missing comparisons)**. We would highly appreciate it if the reviewer could point out any specific method that we have missed.
>
> *W5 (Especially for the missing modality experiments, only 1 comparison is done (against Ma et al., 2022) **.
> We presented missing modality results across 4 datasets, compared with two-branch network, ViLT and Ma et al. Kindly see Tables 6, 7, 8.
> In addition, we provided extensive analysis including tnse on missing modalities scenarios.
> Moreover, we provided a comparison with a latest work on missing modalities scenario during training (Lee et al.).
>
> Comparison of Gazelle with  ViLT and Lee et al. on UPMC Food-$101$, and Hateful Memes under different training and testing settings.  Best results in each setting are shown in bold.
>
> |     Datasets    | Training |      | Testing |      |  Gazelle |  ViLT |  Lee et al. |
> |:---------------:|:--------:|:----:|:-------:|------|----------|-------|-------------|
> |                 | Image    | Text | Image   | Text |          |       |             |
> |   UPMC Food-101 |    100   |  100 |   100   |  100 |   **94.6**   |  91.9 |    92.0 |
> |                 |    100   |  30  |   100   |  30  |   **84.8**   |  66.3 |     74.5    |
> |                 |    30    |  100 |    30   |  100 |   80.0   |  76.7 |     **86.2**    |
> |   Hateful Memes |    100   |  100 |   100   |  100 |   **72.5**   |  70.2 |     71.0    |
> |                 |    100   |  30  |   100   |  30  |   **72.0**   |  60.8 |     59.1    |
> |                 | 30       |  100 |    30   |  100 |   58.8   |  61.6 |     **63.1**    |

---

> ### Comment · Reviewer_q4t6 · 2023-11-17
>
> I thank the authors for reading my comments and providing the rebuttal.
>
> Apologies for the oversight regarding the missing refs (my "find" option on pdfs isn't working properly). They are indeed in the paper. I will update my score from "strong reject" to "reject" in light of this.
>
> The authors did not respond to my question: "Are there no other [modality switching] options? why not set the ratio as a hyperparameter and optimize it?" In essence, the design choices seem arbitrary - they should be better justified, learned, and/or exhaustively tuned.
>
> I also still have concerns, especially regarding the description of the proposed method. I agree with Reviewer ZSk2 that in the absence of a clear theoretical framework for why this should result in better training, the approach can be considered a training trick in its current format. This can be alleviated by providing some theoretical work on why the proposed approach results in a more discriminative joint embedding space. Alternatively, the method can be combined with a few other tricks/small contributions to have a more substantial impact and become a more coherent framework.

---

### Official Review · Reviewer_fm8J · 2023-11-01

**Soundness:** 2 fair
**Presentation:** 3 good
**Contribution:** 2 fair
**Rating:** 5
**Confidence:** 5

**Summary:**

This paper proposes a robust multimodal classification system, which is less susceptible to missing modalities. This system leverages a single-branch network to share weights across multiple modalities, and introduces a novel training scheme for modality switch over input embeddings. Extensive experiments demonstrate the effectiveness of the proposed system.

**Strengths:**

1. The paper is clearly written and contains sufficient details and thorough descriptions of the experimental design.
2. Extensive experiments are conducted to verify the effectiveness of the proposed method.

**Weaknesses:**

1. While ViLT is a good baseline, it is not a "SOTA" method as there are many more advanced models in recent years. Choosing ViLT as the baseline makes the comparison less convincing. Especially, the proposed system uses pre-extracted embeddings (e.g., CLIP).

2. For the table 2-5, the choices of baselines are a little bit out-of-date. The improvements are marginal while the proposed model uses better features with a lot of heuristic designs.

**Questions:**

See the weakness

---

> ### Author Response · Authors · 2023-11-17
> **Rebuttal - fm8J**
>
> **W1 (Baseline)**. We politely disagree as ViLT is a popular multimodal Transformer (Ma et al., 2022). It is evident from the high performances it achieves on complete set of modalities on various multimodal datasets .  Moreover, considering ViLT as baseline highlights that even vision and language Transformers are prone to a drop in performance due to missing modalities.
> We do have more comparisons available in the results section on complete modalities as well as missing modalities during testing.
> Moreover, we provided a comparison with a recently published CVPR 2023 paper (Lee et al. , 2023) (Table below) on missing modalities during training.
> We would highly appreciate it if the reviewer could specify which particular missing modality methods we have missed to provide comparisons with.
>
>
>
> Comparison of Gazelle with  ViLT and Lee et al. on UPMC Food-$101$, and Hateful Memes under different training and testing settings.  Best results in each setting are shown in bold.
>
> |     Datasets    | Training |      | Testing |      |  Gazelle |  ViLT |  Lee et al. |
> |:---------------:|:--------:|:----:|:-------:|------|----------|-------|-------------|
> |                 | Image    | Text | Image   | Text |          |       |             |
> |   UPMC Food-101 |    100   |  100 |   100   |  100 |   **94.6**   |  91.9 |    92.0 |
> |                 |    100   |  30  |   100   |  30  |   **84.8**   |  66.3 |     74.5    |
> |                 |    30    |  100 |    30   |  100 |   80.0   |  76.7 |     **86.2**    |
> |   Hateful Memes |    100   |  100 |   100   |  100 |   **72.5**   |  70.2 |     71.0    |
> |                 |    100   |  30  |   100   |  30  |   **72.0**   |  60.8 |     59.1    |
> |                 | 30       |  100 |    30   |  100 |   58.8   |  61.6 |     **63.1**    |
>
>
> **W2 Heuristic designs**.  In a nutshell, our proposed approach consists of a single-branch with a modality switching mechanism. We are unsure what the reviewer means by **a lot of heuristic design**. We would appreciate it if the reviewer can point out something specific.
> Baseline unimodal image only performance with CLIP on UPMC Food-101 is 81.7. And multimodal performance with Gazelle is 94.6 which is 12.9\% higher than CLIP unimodal image only. Similarly, the baseline unimodal image-only performance with CLIP on Hateful Memes is 68.2. And multimodal performance with Gazelle is 72.5 which is 4.3\% higher than CLIP unimodal image only. We have provided a comprehensive comparison in Tables 2-5.
> We highly appreciate it if the reviewer could specify which particular method we missed to provide a comparison with complete modalities in Tables 2-5.

---

### Official Review · Reviewer_ZSk2 · 2023-11-01

**Soundness:** 2 fair
**Presentation:** 2 fair
**Contribution:** 1 poor
**Rating:** 3
**Confidence:** 4

**Summary:**

The paper presents Gazelle, a simple yet robust multimodal classification model for handling incomplete modalities. The key idea of the model is to use a modality switching mechanism to sequence the embedding streams of single-branch networks. While the experiments demonstrate Gazelle's superior performance in dealing with missing modalities, the paper could benefit from improvements in presentation clarity, additional theoretical analysis, and more robust experimental results.

**Strengths:**

1. The paper introduces a simple yet robust method for handling missing modalities. It is presented in an easy-to-follow manner.
2. The method demonstrates superior robustness when compared to existing state-of-the-art methods.

**Weaknesses:**

1. Incomplete modality/view learning is an important topic in machine learning community, which has achieved great progress in recent years. The authors need to provide a more comprehensive review of the topic.
2. What is the intuition of presenting the modality switching mechanism? A clearer motivation is needed.
3. The proposed method seems to be treated as a training trick. As a general framework, it would be better to provide a theoretical analysis for Gazelle.
4. The readers would be confused with the presentation of Figure 2. For example, what is the mean of each column in S-1, -2, and -3?
5. Can the proposed method handle missing modality in the training stage? How does the method fuse different modalities?
6. The experiment part could be improved by providing a more in-depth analysis. For example, trying to explain why the proposed modality switching strategy is helpful, and whether existing multimodal learning methods benefit from the strategy.


1. In the field of incomplete modality/view learning, it is imperative to provide a comprehensive review of recent advancements within the machine learning community.
2. It would greatly benefit the paper to clarify the intuition behind presenting the modality switching mechanism. A clearer motivation for its inclusion is necessary.
3. The proposed modality switching mechanism can be treated as a training trick. It would be better to provide a theoretical analysis for it.
4. Clarifications should be provided for the presentation of Figure 2, particularly regarding the meanings of each column in S-1, -2, and -3 to avoid confusion for readers.
5. Further details regarding the capability of the proposed method to handle missing modalities during the training stage and insights into how it effectively fuses different modalities are needed for clarity.
6. The experiment part could be improved by providing a more in-depth analysis. For example, explain how the proposed modality switching strategy improves robustness, and whether existing multimodal learning methods benefit from the strategy.

**Questions:**

please see the weaknesses.

---

> ### Author Response · Authors · 2023-11-17
> **Rebuttal - ZSk2**
>
> **W1 (Review of the topic)**. We have provided a comprehensive overview of related work in Section 2. We would highly appreciate it if the reviewer could provide specific details on any missing references.
>
> **W2 (Intuition)**. In existing methods, the performance deterioration in the case of missing modalities may be attributed to the commonly used multi-branch design.
> Such a design may learn weights in a way that the final performance is highly dependent on the correct combination of input modalities.
> Modality switching mechanism is an important component of our method that enables providing input modalities in a sequential fashion so that a single branch network is able to learn from both modalities without any dependence on complete set of modalities. We propose three modality switches to find the best combination of input modalities.
>
> **W3 (Training trick)**. We disagree that the proposed method is a training trick. It is a different architecture whose training scheme is designed in a way that improves one of the limitation of current SOTA methods.
> Moreover, we have provided detailed empirical studies on four datasets and numerous settings of missing modalities, which establishes the general applicability of our approach towards handling missing modality.
>
> **W4 (Presentation of Fig. 2)**.  Fig. 2 shows the modality switching mechanism and each column in S-1, -2, and -3 represents a *batch* of size 5. More specifically, S-1) all batches are multimodal in all epochs, S-2) half of the batches in each epoch are multimodal and the other half are unimodal, S-3) each batch has a different randomly selected modality. More details are provided in Section 3.2.
>
> **W5 (Training stage)**.  It can handle missing modalities at the training stage.  We provided a comparison with a recently published paper  (Lee et al. , 2023) (Table below) on missing modalities during training.
>
> |     Datasets    | Training |      | Testing |      |  Gazelle |  ViLT |  Lee et al. |
> |:---------------:|:--------:|:----:|:-------:|------|----------|-------|-------------|
> |                 | Image    | Text | Image   | Text |          |       |             |
> |   UPMC Food-101 |    100   |  100 |   100   |  100 |   **94.6**   |  91.9 |    92.0 |
> |                 |    100   |  30  |   100   |  30  |   **84.8**   |  66.3 |     74.5    |
> |                 |    30    |  100 |    30   |  100 |   80.0   |  76.7 |     **86.2**    |
> |   Hateful Memes |    100   |  100 |   100   |  100 |   **72.5**   |  70.2 |     71.0    |
> |                 |    100   |  30  |   100   |  30  |   **72.0**   |  60.8 |     59.1    |
> |                 | 30       |  100 |    30   |  100 |   58.8   |  61.6 |     **63.1**    |
>
> We would like to clarify that we don't use fusion of any sort in our training. As explained in manuscript, only at test time, if complete modalities are present, late fusion is employed by taking an average of the logits obtained from the softmax layer over multiple modalities to compute classification score.
>
> **W6 (In-depth analysis)**.  We appreciate the suggestion.
> It is not trivial to apply our approach directly on ViLT, as the architectures used in these methods require a complete set of modalities for training.  We do provide a comparison with Two-branch model, which is essentially a baseline of our approach without the proposed training scheme. As seen in Tables 7 and 8, our approach consistently demonstrates superior performances compared to Two-branch model on all settings.
> In addition, following your suggestion, we carry out some experiments using emb. extracted from ViT (image modality) \& Doc2Vec ( text modality) which makes our approach comparable with ViLT.  Results are available in the following table
>
> |        Dataset      | Training |      | Testing |      |   Gazelle |   ViLT |
> |:-------------------:|:--------:|:----:|:-------:|:----:|:---------:|:------:|
> |                     |   Image  | Text |  Image  | Text |           |        |
> |       UPMC Food-101 |    100   |  100 |   100   |  100 |    **91.9**   |  88.6  |
> |                     |    100   |  100 |   100   |  90  |   **88.2**   |  86.5  |
> |                     |    100   |  100 |   100   |  70  |    80.7   |  **82.1**  |
> |                     |    100   |  100 |   100   |  50  |    73.3   |  **77.9**  |
> |                     |    100   |  100 |   100   |  30  |    65.9   |  **74.0**  |
> |                     |    100   |  100 |   100   |  10  |    58.4   |  **69.7**  |
> |       Hateful Memes |    100   |  100 |   100   |  100 |    **70.2**   |  61.1  |
> |                     |    100   |  100 |   100   |  90  |    **68.8**   |  60.5  |
> |                     |    100   |  100 |   100   |  70  |    **65.9**   |  60.4  |
> |                     |    100   |  100 |   100   |  50  |    **63.6**   |  59.7  |
> |                     |    100   |  100 |   100   |  30  |    **60.2**   |  59.4  |
> |                     |    100   |  100 |   100   |  10  |    54.9   |  **58.7**  |